# *Epilobium**pyrricholophum* Extract Suppresses Porcine Pancreatic Elastase and Cigarette Smoke Extract-Induced Inflammatory response in a Chronic Obstructive Pulmonary Disease Model

**DOI:** 10.3390/foods10122929

**Published:** 2021-11-26

**Authors:** Sun Young Jung, Gun-Dong Kim, Dae Woon Choi, Dong-Uk Shin, Ji-Eun Eom, Seung Yong Kim, Ok Hee Chai, Hyun-Jin Kim, So-Young Lee, Hee Soon Shin

**Affiliations:** 1Research Division of Food Functionality, Korea Food Research Institute, Wanju 55365, Korea; jsy5528@gmail.com (S.Y.J.); kgd@kfri.re.kr (G.-D.K.); choidw19@gmail.com (D.W.C.); 2Department of Food Biotechnology, University of Science and Technology, Daejeon 34113, Korea; 50010@kfri.re.kr; 3Research Group of Natural Materials and Metabolism, Korea Food Research Institute, Wanju 55365, Korea; 4Food Functional Evaluation Support Team, Korea Food Research Institute, Wanju 55365, Korea; jeeom@kfri.re.kr; 5Department of Food Science and Technology, Jeonbuk National University, Jeonju 54896, Korea; tmd-dyd0828@gmail.com; 6Department of Anatomy, Institute of Medical Science, Jeonbuk National University Medical School, Jeonju 54907, Korea; okchai1004@jbnu.ac.kr; 7Division of Applied Life Science (BK21 Four), Department of Food Science and Technology, and Institute of Agriculture and Life Science, Gyeongsang National University, 501 Jinjudaero, Jinju 52828, Gyeongsangnam-do, Korea; hyunjkim@gnu.ac.kr; 8EZmass. Co. Ltd., 501 Jinjudaero, Jinju 55365, Gyeongsangnam-do, Korea

**Keywords:** chronic obstructive pulmonary disease, cigarette smoke, inflammation, *Epilobium*

## Abstract

Chronic airway exposure to harmful substances, such as deleterious gases, cigarette smoke (CS), and particulate matter, triggers chronic obstructive pulmonary disease (COPD), characterized by impaired lung function and unbridled immune responses. Emerging epigenomic and genomic evidence suggests that excessive recruitment of alveolar macrophages and neutrophils contributes to COPD pathogenesis by producing various inflammatory mediators, such as reactive oxygen species (ROS), neutrophil elastase, interleukin (IL) 6, and IL8. Recent studies showed that *Epilobium* species attenuated ROS, myeloperoxidase, and inflammatory cytokine production in murine and human innate immune cells. Although the *Epilobium* genus exerts anti-inflammatory, antioxidant, and antimicrobial effects, the question of whether the *Epilobium* species regulate lung inflammation and innate immune response in COPD has not been investigated. In this study, *Epilobium pyrricholophum* extract (EPE) suppressed inflammatory cell recruitment and clinical symptoms in porcine pancreatic elastase and CS extract-induced COPD mice. In addition, EPE attenuated inflammatory gene expression by suppressing MAPKs and NFκB activity. Furthermore, UPLC-Q-TOF MS analyses revealed the anti-inflammatory effects of the identified phytochemical constituents of EPE. Collectively, our studies revealed that EPE represses the innate immune response and inflammatory gene expression in COPD pathogenesis in mice. These findings provide insights into new therapeutic approaches for treating COPD.

## 1. Introduction

Chronic obstructive pulmonary disease (COPD), one of the leading causes of mortality globally, is induced and developed by repetitive airway exposure to harmful particles, such as cigarette smoke and particulate matter [1,2]. COPD is characterized by shortness of breath, airflow limitation, inflammatory airway obstruction, emphysema, and bronchitis, leading to impaired lung function [3]. Although COPD prevalence has gradually increased, existing therapeutic applications are limited to ameliorating the severity and frequency of COPD exacerbations.

Increased recruitment of inflammatory cells, such as alveolar macrophages and neutrophils in the lungs, plays an important role in disease progression and COPD severity [4,5]. A recent study showed that activated neutrophils recruited to the lungs secrete various inflammatory mediators, including reactive oxygen species (ROS) and lytic enzymes, particularly CD63^+^/CD66b^+^ exosomes containing neutrophil elastase, leading to emphysema [4,6]. Similarly, an influx of alveolar macrophages to the airway and lungs under COPD pathogenesis revealed elevated production of cytokines, chemokines, reactive nitrogen, and oxygen species [5]. Therefore, suppression of the unbridled immune response driven by activated macrophages and neutrophils may apply to COPD treatment.

A recent study showed that *Epilobium* species, including *E. angustifolium*, *E. hirsutum*, and *E. parviflorum*, suppress myeloperoxidase, leukotriene B4, and ROS production in human neutrophils [7]. Notably, *E. angustifolium* extract attenuates ROS production, nuclear factor kappa-light-chain-enhancer of activated B cells (NFκB)-p65 activity, cell motility, and inflammatory cytokine production, including IFNγ, IL1β, TNF, IL6, and IL8 in humans and murine innate immune cells, such as neutrophils and macrophages [8].

The genus *Epilobium* belongs to the Onagraceae family. It consists of over 200 species widely distributed worldwide and is used as a traditional remedy for diseases such as benign prostate hyperplasia, prostatitis, urinary tract associated disease, and gastrointestinal tract diseases [9,10]. Pharmacological analyses of the *Epilobium* species revealed anti-inflammatory, anti-androgenic, antioxidant, anti-tumor, and analgesic functions with a composition of various phytochemical compounds, including flavonoids, phenolic acids, tannins, steroids, and fatty acids [9,10]. However, whether the *Epilobium* genus regulates lung inflammation and innate immune cell-mediated pathogenic responses by modulating inflammatory gene programming in COPD has not been investigated.

In this study, we analyzed the phytochemical composition of *E. pyrricholophum*, which belongs to the Onagraceae family distributed in Asian countries, including Korea, China, and Japan, using ultra-performance liquid chromatography quadrupole time of flight (UPLC-Q-TOF) MS [11]. Notably, corilagin, which was identified as a phytochemical constituent of *E. pyrricholophum*, significantly attenuated CSE-induced IL6 and IL8 expression in vitro. Furthermore, we demonstrate that *E*. *pyrricholophum* significantly attenuates pro-inflammatory gene expression and the recruitment of immune cells to the lungs after porcine pancreatic elastase and cigarette smoke extract (CSE) exposure in mice. Our gene expression analysis studies revealed that *E*. *pyrricholophum* suppresses CSE-induced IL6 and IL8 gene expression by diminishing mitogen-activated protein kinase and the NFκB activity. Based on our observations, we propose that *E*. *pyrricholophum* has potential therapeutic benefits in COPD treatment.

## 2. Materials and Methods

### 2.1. Materials

RPMI 1640, penicillin-streptomycin, and phosphate-buffered saline were obtained from Welgene (Daegu, Korea). Fetal bovine serum was purchased from Biowest (Nuaillé, France). The mitogen-activated protein kinase/extracellular signal-regulated kinase inhibitor U0126 was obtained from Promega (Madison, WI, USA). The p38 inhibitor SB203580 and myricitrin (9125) were purchased from Sigma Aldrich (St. Louis, MO, USA). Quercitrin was obtained from Cayman Chemical (19866; Ann Arbor, MI, USA). Kaempferol 3-rhamnoside (PHL83864), corilagin (G0424), and avicularin (44006) were purchased from Merck & Co., Inc. (Kenilworth, NJ, USA). Myricetin 3-O-galactoside (CFN97817) was obtained from ChemFaces (Wuhan, China). Brevifolin carboxylic acid was purchased from BOC Sciences (18490-95-7; Shirley, NY, USA). Water-soluble tetrazolium salt was obtained from GE Vitek (WST-1; Seoul, Korea).

### 2.2. Preparation of Plant Material

*E*. *pyrricholophum* (033-040), harvested in 2008 by Bonghwa-gun, Gyeongsangbuk-do, Korea, was purchased from the Korea Plant Extract Bank at the Korea Research Institute of Bioscience and Biotechnology. The dried plant (20 g) was crushed and dissolved in 1 L HPLC grade 99.9% methyl alcohol. The methanol extract was ultrasonicated at room temperature for 2 h and filtered using a cartridge filter. The filtrate was evaporated and obtained 2.17 g of 10.85% yield powder which was dissolved in normal saline for subsequent oral usage in mice.

### 2.3. Preparation of Cigarette Smoke Extract (CSE)

The 3R4F reference cigarettes were obtained from the University of Kentucky. Filter-removed cigarettes were combusted and passed through 10 mL of PBS with a constant airflow using a vacuum pump for 2 min. After filtration to remove residual particles and bacteria, the optical density of the CSE was measured at 320 nm for standardization. The extract was prepared fresh before each experiment and instilled after adjusting the pH between 7.00 and 7.40 with an optical density between 0.90 to 1.00 at 320 nm.

### 2.4. Cell Culture

NCI-H292 human airway epithelial cells were cultured in RPMI 1640 medium supplemented with 10% fetal bovine serum, 100 U/mL penicillin, 100 μg/mL streptomycin, 4500 mg/mL D-glucose, 2 mM L-glutamine, 10 mM HEPES, 1 mM sodium pyruvate, and 1500 mg/mL sodium bicarbonate in a humidified incubator (5% CO_2_ and 37 °C).

### 2.5. Animal Study

Animal studies were conducted following institutional and national guidelines, and all animal procedures were approved by the Korea Food Research Institute Animal Care and Use Committee (KFRI-M-18030). All the mice were bred and maintained under pathogen-free conditions at 23 ± 2 °C and a relative humidity of 50 ± 10% and maintained on a 12 h light/dark cycle. Male BALB/c mice were obtained from OrietBio Inc. (Gwangju, Korea), and, following a one-week acclimatization period, the mice were randomly divided into four groups (*n* = 10).

The COPD model was established by intranasal delivery of porcine pancreatic elastase (PPE; EMD Millipore Corporation, Burlington, MA, USA) with CSE. Briefly, the mice received 1.2 unit/20 μL PPE intranasally on days 8 and 14 and were exposed to 20 μL of 2% CSE on days 8, 10, 12, 14, 16, and 18 after being anesthetized with isoflurane. *Epilobium pyrricholophum* extract (EPE, 200 mg/kg) was administered daily from day 1 to 19. Roflumilast (10 mg/kg, Sigma Aldrich, St. Louis, MO, USA) was administered from days 8 to 19 as a positive control. The naive group that received the saline served as vehicle control. These mice were euthanized on day 19 and intubated with 1 mL PBS with 1% FBS through the tracheal cannula to collect bronchoalveolar lavage fluid (BALF). The total cell count was measured using an ADAM-MC™ automatic cell counter (NanoEntek, Seoul, Korea), and the collected BALF was centrifuged to separate the cellular and liquid components for further analyses.

In parallel studies, 250 μL BALF was applied to coated cytoslide (SHANDON; Thermo Scientific, Waltham, MA, USA) using a cytospin device centrifuge (5403; Eppendorf, Hamburg, Germany) at 1000 rpm for 10 min, and the cells were stained using Diff Quick reagent (38721; SYSMEX, Kobe, Japan). The production of chemokines and cytokines, such as keratinocyte-derived chemokine (KC); monocyte chemoattractant protein (MCP) 1; macrophage-derived chemokine (MDC); thymus and activation regulated chemokine (TARC); macrophage inflammatory protein (MIP) 2; tumor necrosis factor (TNF); interleukin (IL) 1β; IL6; and IL8 in the BALF, were assessed using a Q-plex assay kit (Quansys Biosciences, Logan, UT, USA). Briefly, solutions for the calibration curve and the samples diluted 1:2 with sample diluent were added to the Q-plex^TM^ Array 96-well plate and shaken for 1 h at RT. After washing, the Detection Mix was added to the plate and shaken for 1 h at RT. Streptavidin-HRP was added to the plate and shaken for 15 min and then substrate solutions were added. To measure the intensity of the spot response, the image of the plate was captured using a Q-View^TM^ Imager Pro (Quansys Biosciences, Logan, UT, USA). Excised lung tissues were fixed in 10% buffered formalin, embedded in paraffin, stained with hematoxylin, eosin, and periodic acid-Schiff (Sigma Aldrich, St. Louis, MO, USA). Images were acquired using a microscope and quantified using ImageJ software version 1.53k (NIH, Bethesda, MD, USA; http://imagej.nih.gov/ij, accessed on 15 September 2021).

### 2.6. RNA Extraction, Real-time Quantitative PCR, and Western Blotting

Total RNA was isolated from the indicated samples using the QIAzol^®^ Lysis Reagent kit (QIAGEN, Valencia, CA, USA). One microgram of total RNA was reverse-transcribed using M-MulV reverse transcriptase in the presence of random hexamers and oligo (dT) primers. Real-time quantitative PCR was performed using Rotor-Gene SYBR Green master mix (QIAGEN, Valencia, CA, USA) on a Rotor-Gene Q real-time PCR system in the presence of gene-specific primers. The primer sequences were as follows: IL6 forward, 5′-ATGAACTCCTTCTCCACAAGC-3′ and reverse, 5′-TGGACTGCAGGAACTCCTT-3′; IL8 forward, 5′-ATGACTTCCAAGCTGGCCGTGGCT-3′ and reverse, 5′-TCTCAGCCCTCTTCAAAAACTTCTC-3′; GAPDH forward, 5′-TGCACCACCAACTGCTTA-3′ and reverse, 5′-GGCATGGACTGTGGTCAT-3′.

The cellular protein extracts were subjected to Western blot analysis, as described below. The cells were lysed using radioimmunoprecipitation lysis buffer (Cell Signaling Technology, Danvers, MA, USA) supplemented with a protease inhibitor cocktail (Thermo Fisher Scientific, Waltham, MA, USA) following the specified treatments. Equal quantities of total protein were incubated with primary antibodies, including ERK (4695S), p-ERK (4370S), p38 (9212S), p-p38 (9211S), NFκB-p65 (8242S), p-NFκB-p65 (3033S; Cell Signaling Technology, Danvers, MA, USA), and β-actin (D0615; Santa Cruz Biotechnology, CA, USA), using an automated capillary-based size sorting system (WES; ProteinSimple, Santa Clara, CA, USA). Protein analyses were performed using the Compass software version 6.0 (ProteinSimple, San Jose, CA, USA).

### 2.7. UPLC-Q-TOF MS Analysis

The bioactive compounds in the EPE were analyzed using a UPLC-Q-TOF MS system (Waters, Milford, MA, USA). The metabolites were separated using an Acquity UPLC BEH C_18_ column (2.1 mm × 100 mm, 1.7 m; Waters Corp., Milford, MA, USA) equilibrated with water containing 0.1% formic acid (solvent A) and eluted with a gradient of acetonitrile containing 0.1% formic acid (solvent B). Eluted compounds were detected from an *m/z* 50 to 1500 range with a scan time of 0.2 sec and an interscan delay time of 0.02 sec using the Q-TOF MS system with positive/negative electrospray ionization mode using the following instrument settings: capillary voltage, 3.0/2.5 kV; sample cone voltage, 30/20 V; ion source temperature, 100 °C; desolvation temperature, 400 °C; cone gas flow rate, 30 L/h; and desolvation gas flow rate, 800/900 L/h. All analyses were performed using a lock spray to ensure accuracy and reproducibility. Leucine-enkephalin ([M + H] = 556.2771 Da; [M − H] = 554.2615 Da), used as lock mass, was infused at a flow rate of 20 μL/min and a frequency of 10 s. MS/MS spectra were obtained using a collision energy ramp from 10 to 30 eV. LC/MS data, including retention time, *m/z*, and ion intensity, were extracted using UNIFI version 1.8.2.169 (Waters) and assembled into a data matrix. The compounds were tentatively identified based on online databases, including the ChemSpider database in UNIFI and the METLIN database (www.metlin.scripps.edu, accessed on 7 September 2021).

### 2.8. Quantification and Statistical Analysis

All data, unless indicated, are presented as mean ± standard deviation (SD). The statistical significance of differences between the two groups was analyzed by Student’s *t*-test, one-way analysis of variance (ANOVA), or two-way ANOVA with F-protected Fisher’s least significant difference tests. The statistical significance was set at *p* < 0.05.

## 3. Results

### 3.1. EPE Attenuates Inflammatory Cell Infiltration and Pro-Inflammatory Gene Expression in COPD Mouse BALF

Previous studies have reported that *Epilobium* extract has beneficial effects, including anti-inflammatory, anti-tumor, antimicrobial, antioxidative, and analgesic effects [12,13,14,15,16]. Therefore, we hypothesized that EPE could be used as a therapeutic agent for inflammatory and allergic diseases. We utilized PPE and CSE to test this hypothesis in order to establish a COPD model (Figure 1A). Accordingly, oral administration of EPE was initiated 7 days before PPE and CSE induction to identify both the prophylactic and the therapeutic effects. Because the phosphodiesterase 4 inhibitor roflumilast (ROF) has been used to maintain COPD exacerbations, we used ROF as a positive control to compare EPE treatment [17]. As shown in Figure 1B, mice challenged with PPE and CSE exhibited significantly increased neutrophil infiltration in the BALF. Consistently, our analyses showed that the recruitment of macrophages, neutrophils, and lymphocytes to the BALF was significantly elevated in COPD mice (Figure 1C). However, elevated inflammatory cell recruitment, including macrophages, neutrophils, and lymphocytes in the BALF, was decreased in mice treated with ROF (Figure 1B,C). Interestingly, mice treated with EPE displayed a dramatically attenuated number of total cells as well as the recruitment of macrophages, neutrophils, and lymphocytes in the BALF, similar to the mice treated with ROF (Figure 1B,C). Next, we examined whether EPE treatment altered inflammatory gene expression in the BALF of COPD mice. Our studies showed that chemokine (KC, MCP1, MDC, TARC, and MIP2) and inflammatory cytokine (TNF, IL1β, and IL6) production was highly elevated in the BALF of PPE and CSE-elicited COPD mice compared to the naive mice (Figure 1D). However, heightened pro-inflammatory chemokine and cytokine levels in the BALF were significantly attenuated in the EPE-treated mice, similar to the ROF-treated mice (Figure 1D). Our results collectively demonstrate that EPE significantly attenuates inflammatory gene expression and inflammatory cell infiltration in the BALF of PPE- and CSE-induced COPD mice.

### 3.2. EPE Ameliorates PPE and CSE-Induced Lung Inflammation in COPD Mice

Concurrently with BALF analysis, we examined whether EPE treatment attenuated pro-inflammatory gene expression and inflammatory cell recruitment to the lungs following the PPE and CSE challenge. Accordingly, lung tissues from individual groups of mice were subjected to detailed histopathological evaluations involving hematoxylin and eosin (H&E) staining and periodic acid-Schiff (PAS) staining. As anticipated, PPE-and CSE-induced histological alterations exhibited various clinical symptoms, such as highly elevated inflammatory cell recruitment, disruption of elastic fibers, increases in the airway wall and bronchiole thickness, epithelial cell hyperplasia, and hypersecretion of airway mucus, in COPD mice (Figure 2A). Remarkably, H&E and PAS staining showed that EPE-treated mice exhibited significantly attenuated clinical symptoms following the PPE and CSE challenge, similar to the mice that received ROF (Figure 2A). Next, we evaluated whether EPE treatment suppressed inflammatory gene expression in the lungs of COPD mice. As shown in Figure 2B, the PPE and CSE exposure markedly induced inflammatory chemokine and cytokine production involving KC, MCP1, MDC, TARC, MIP2, TNF, IL1β, and IL6 compared to that of the naive mice. However, the mice that received EPE exhibited significantly suppressed inflammatory gene expression in the lung tissues following PPE and CSE exposure, consistent with the observations in the BALF and immunohistochemistry analyses (Figure 2B). ROF treatment reduced PPE- and CSE-induced KC and IL1β levels in the lungs, whereas the levels of MCP1, MDC, TARC, MIP2, TNF, and IL6 were increased, unexpectedly (Figure 2B). Although phosphodiesterase 4 inhibitors are used as a therapeutic application for airway inflammation, ROF has been shown to cause an unknown etiology in experimental models. Notably, 100 mg/kg ROF induced increases in the number of neutrophils and KC levels in the plasma and the lung tissues of the mouse model [18]. Furthermore, Kasetty et al. [19] demonstrated that ROF increases the plasma levels of IL6, MCP1, and TNF while suppressing the infiltration of neutrophils in the BALF after bacterial infection, similar to our observations. Therefore, we are conducting further studies to investigate the appropriate dosage and molecular mechanisms involved in that ROF treatment in the elastolytic enzyme and CSE-induced COPD model. Collectively, our results demonstrate that EPE restrains PPE and CSE-elicited lung inflammation in COPD mice.

### 3.3. EPE Suppresses CSE-Induced Inflammatory Gene Expression through the MAPK and NFκB Pathway in NCI-H292 Cells

The human pulmonary mucoepidermoid cell line, NCI-H292, has been used to establish an airway inflammation model in vitro to elucidate intracellular signaling pathways, including airway inflammatory gene expression and mucin secretion [20,21]. Previous studies have reported that highly upregulated IL6 and IL8 gene expression in the lungs is a typical characteristic of COPD [22,23]. Therefore, we examined whether EPE treatment altered CSE-induced inflammatory gene expression in NCI-H292 cells. NCI-H292 cells were stimulated with 2% CSE and RT-qPCR and ELISA assessed the gene expression of IL6 and IL8 to evaluate this notion. As shown in Figure 3A, CSE exposure significantly induced IL6 and IL8 mRNA expression in the NCI-H292 cells. Interestingly, EPE treatment markedly downregulated IL6 and IL8 mRNA expression in a concentration-dependent manner (Figure 3A). Accordingly, IL6 and IL8 production evaluation in supernatants was measured by ELISA following CSE exposure (Figure 3B). Consistent with the mRNA expression analyses, EPE treatment significantly suppressed CSE-induced IL6 and IL8 production in NCI-H292 cells (Figure 3B). Recent studies have reported that CSE-elicited inflammatory gene expression, such as IL6 and IL8, is dependent on the phosphorylation of mitogen-activated protein kinases (MAPKs), such as p38 and ERK, as well as the activation of NFκB [24,25]. Accordingly, to investigate the signaling pathway involved in CSE-induced IL6 and IL8 production in NCI-H292 cells, we evaluated the phosphorylation levels of MAPKs and NFκB-p65 by Western blotting. NCI-H292 cells were stimulated with 2% CSE, and total protein extracts were examined for ERK, p38, and NFκB-p65 phosphorylation (Figure 3C,D). As anticipated, CSE exposure significantly elevated the ERK, p38 (Figure 3C), and NFκB-p65 phosphorylation levels (Figure 3D) in the NCI-H292 cells. Intriguingly, EPE 5 μg/mL treatment significantly suppressed the phosphorylation levels of ERK, p38, and NFκB-p65 in the NCI-H292 cells following the 2% CSE exposure (Figure 3C,D). Next, we investigated whether EPE treatment could diminish IL6 and IL8 expression through the ERK and p38 signaling pathways in NCI-H292 cells following CSE exposure. Accordingly, we utilized pharmacological ERK and p38 inhibitors, U0126 and SB203580, in the absence or presence of EPE treatment. These cells were stimulated with 2% CSE, and ELISA evaluated the expression of IL6 and IL8. As anticipated, the CSE-induced IL6 and IL8 productions were significantly downregulated by U0126 and SB203580 treatment (Figure 3E,F). Remarkably, EPE treatment significantly attenuated IL6 and IL8 production in the NCI-H292 cells following CSE exposure, similar to the pharmacological blockade of the ERK and p38 pathways (Figure 3E,F). In addition, the combination of EPE and either U0126 or SB20358 treatment exhibited significantly lower levels of IL6 and IL8 compared to a single treatment of EPE or each inhibitor in the NCI-H292 cells following CSE exposure (Figure 3E,F). Taken together, our analyses demonstrated that EPE suppresses CSE-induced IL6 and IL8 production through the inhibition of ERK, p38, and NFκB-p65 phosphorylation in NCI-H292 cells.

### 3.4. Identification of the Active Compounds in EPE That Ameliorates Inflammatory Gene Expression

The phytochemical constituents of EPE were analyzed using UPLC-Q-TOF-MS with their retention time, molecular weight, fragment ion, common name, and molecular formula. Seven identified compounds (myricetin 3-O-galactoside, quercitrin, corilagin, brevifolin carboxylic acid, myricitrin, avicularin, and kaempferol 3-rhamnoside) are reported in Table 1, and the associated chromatograms are presented in Figure 4A. Peaks 1 and 2 were identified as myricetin 3-O-galactoside and quercitrin (R_T_ = 3.66 min and 4.05 min) at *m*/*z* 481 and 447 in the MS spectrum in ESI+ mode. Each of the respective peaks of the fragment ions were observed at *m*/*z* 153, 303, 319, and 287, 303, 305, and 435. Peak 3 (R_T_ = 2.70 min) was recognized as corilagin at *m*/*z* 633 in the MS spectrum of the ESI– mode, and peaks of the fragment ions were identified at *m*/*z* 231, 301, 633, and 634, respectively. Peaks 4, 5, 6, and 7 (R_T_ = 3.34, 3.88, 4.16, and 4.38 min) were characterized as brevifolin carboxylic acid, myricitrin, avicularin, and kaempferol 3-rhamnoside at *m*/*z* 291, 463, 449, and 431 in the MS spectrum of the ESI mode, respectively. A recent study showed that polyphenols are major constituents of *Epilobium* species, including flavonoids, phenolic acids, and tannins, and possess multiple biological functions, including anti-inflammation, anti-oxidation, anti-infection, and anti-proliferation [9]. Therefore, we examined whether the identified flavonoids (myricitrin, quercitrin, kaempferol 3-rhamnoside, myricetin 3-O-galactoside, and avicularin) and polyphenols (corilagin and brevifolin carboxylic acid) altered CSE-induced inflammatory gene expression in the NCI-H292 cells. The NCI-H292 cells were stimulated with 2% CSE in the presence or absence of selectively identified compounds, and ELISA assessed the gene expression of IL6 and IL8 to evaluate this hypothesis. Table 2 and Table 3 show that 2% CSE exposure significantly induced IL6 and IL8 expression in the NCI-H292 cells. Accordingly, myricetin 3-O-galactoside, avicularin, corilagin, and brevifolin carboxylic acid treatments downregulated IL6 expression in a concentration-dependent manner (Table 2). In addition, at a 10 mM concentration of quercitrin and myricitrin, a 30 mM and 50 mM treatment showed suppressive effects of IL6 following CSE treatment in the NCI-H292 cells (Table 2). However, most compounds failed to attenuate CSE-induced IL8 expression in the NCI-H292 cells (Table 3). Interestingly, among the compounds selectively identified from EPE, only corilagin significantly attenuated CSE-induced IL6 and IL8 expression in the NCI-H292 cells (Figure 4B,C). Collectively, our results demonstrate that the polyphenol constituents, corilagin, exhibit remarkable anti-inflammatory functions among the identified phytochemical compounds from EPE using UPLC-Q-TOF-MS.

## 4. Discussion

Our study is the first to show that the aqueous methanol extract of *E*. *pyrricholophum* represses PPE- and CSE-induced lung inflammation in a murine COPD model. The key observations of our study are as follows: EPE suppresses elevated IL6 and IL8 mRNA and protein expression following CSE exposure in human airway epithelial NCI-H292 cells; EPE attenuates CSE-induced IL6 and IL8 expression which depends on both MAPKs (ERK and p38) and NFκB-p65 signaling pathways in NCI-H292 cells; EPE ameliorates the infiltration of inflammatory cells, including macrophages, neutrophils, and lymphocytes in BALF of PPE and the CSE-induced COPD mouse model; EPE significantly suppresses inflammatory gene expression, including KC, MCP1, MDC, TARC, MIP2, TNF, IL1β, and IL6 in both the BALF and the lung tissues of the murine COPD model following PPE and CSE exposure; EPE attenuates PPE and CSE-induced COPD-like clinical symptoms, such as elastic fiber disruption, airway wall, and bronchiole thickness increases, epithelial cell hyperplasia, and mucus hypersecretion in the lung tissues of the mouse COPD model; in particular, phytochemical constituents, corilagin, from EPE significantly represses CSE-induced IL6 and IL8 expression in a concentration-dependent manner in NCI-H292 cells. Collectively, our observations demonstrate that EPE suppresses the immune response and inflammatory gene expression in a murine COPD model.

COPD is a complex chronic inflammatory disease associated with multiple etiological factors, such as inflammatory response, bacterial infection, oxidative stress, cilia dysfunction, long-term exposure to dust or particulate matter, and cigarette smoking (CS) [26]. Recent studies have shown that CS is correlated with the activation and infiltration of inflammatory cells, including macrophages, dendritic cells, neutrophils, and pro-inflammatory cytokine production, such as IL1β, IL6, IL8, and TNF [27,28]. Accordingly, smoking COPD patients showed a significantly lower forced expiratory volume in 1 sec (FEV_1_) and FEV_1_/forced vital capacity ratio than both the COPD patients exposed to biomass smoke and the healthy control group. In contrast, the smoking COPD patients had increased neutrophil and lymphocyte numbers and higher C-reactive protein blood levels than the control group [29]. Furthermore, a recent study reported that long-term CSE exposure significantly upregulated pro-inflammatory gene expression levels, including IL1β, IL6, and IL8 in the human bronchial epithelial cell line BEAS-2B from COPD GOLD stage IV patients compared to the never-smoking controls [30]. Concordant with these observations, our current study revealed that CSE and PPE exposure heightened small airway obstruction and fibrotic changes in the lung parenchyma through exacerbated pro-inflammatory gene expression and infiltration of inflammatory cells, including macrophages, neutrophils, and lymphocytes, to the lungs.

Persistent neutrophilic inflammation induced by bacterial infection, dysregulated innate immune response, and CS are critical in COPD [31,32]. Activated neutrophils enhance the release of neutrophil extracellular trap (NET) formation, involving multiple inflammatory mediators and proteases, such as NE, myeloperoxidase, and matrix metalloproteinase (MMP) [33]. Previous studies reported that the dominant pulmonary NE inhibitor, alpha1-antitrypsin (α1AT)-deficient patients exhibit rapidly progressive emphysematous-like conditions and early onset emphysema [34,35]. Notably, NE derived from CD63^+^/CD66b^+^ exosomes from activated neutrophils induces the degradation of the pulmonary interstitial matrix, composed of extracellular matrix proteins, collagens, and elastin structure, and has resistance against α1AT, consequently triggering pulmonary architectural distortion and inflammatory airway obstruction [6]. Consistently, our current studies have identified that either EPE or ROF treatment attenuates lung inflammation and matrix remodeling by suppressing the infiltration of neutrophils in the BALF and lungs following PPE and CSE exposure in an in vivo COPD model. Given these observations, defining the regulation mechanisms of the unbridled activity of neutrophil and protease interplay between inflammation and matrix remodeling may contribute to alternative therapeutic approaches to COPD or bronchopulmonary dysplasia.

Alveolar macrophages also contribute to chronic inflammation, particularly in response to long-term CS exposure in patients with COPD [5]. Accordingly, COPD patients revealed a significantly increased number of macrophages in BALF, sputum, airways, and lung parenchyma associated with the severity of emphysema [36]. Activated alveolar macrophages by CSE exposure produce pro-inflammatory genes, including TNF, CXCL1, IL8, and MCP1, and elastolytic enzymes, such as MMP2, MMP9, and MMP12 [32]. MMP9 is a major elastolytic enzyme produced by alveolar macrophages in COPD patients and cigarette smokers [37,38]. Intriguingly, elevated levels of MMP9 in sputum and plasma caused emphysema, as well as diminished FEV_1_, carbon monoxide transfer factor, and oxygen saturation in COPD exacerbation in patients exposed to CS [39,40]. Consistent with these observations, MMP12-deficient mice repressed the influx of alveolar macrophages to the lungs and the development of emphysema following long-term CS exposure [41]. In this context, identifying transcriptional regulation correlated with inducible proteolytic enzymes and inflammatory gene expression in inflammatory cells, such as neutrophils, macrophages, lymphocytes, and epithelial cells, might resolve excessive inflammatory responses and tissue remodeling in patients with COPD.

ROS mediated by CS-induced oxidative stress activates the p38 MAPK and NFκB signaling pathways in both the epithelial cells and the alveolar macrophages of COPD patients and enhances the expression of inflammatory mediators, such as IL8, TNF, and the granulocyte-macrophage colony-stimulating factor [42,43]. These robust pro-inflammatory milieus consequently amplify inflammatory responses in patients with COPD. In addition, ROS contributes to elastolytic enzyme production and airway mucus obstruction, which leads to airflow limitation and accelerates tissue damage [44]. A previous study reported that IKTA mice, which constitutively express the human inhibitor of nuclear factor kappa-B kinase subunit β in the airway epithelium, showed early postnatal death with hypoxemia and disrupted elastic fiber formation [45]. Based on these observations, we hypothesized that inhibition of the inflammatory response mediated by MAPKs and NFκB might attenuate pro-inflammatory gene expression in the human pulmonary epithelium following CSE exposure. Consistently, our current studies showed that EPE significantly repressed CSE-induced IL6 and IL8 mRNA and protein expression in human pulmonary epithelial NCI-H292 cells via the suppressive effects of ERK, p38, and the NFκB-p65 signaling pathway, respectively.

Inhaled bronchodilators and corticosteroids have been widely used to manage COPD and COPD exacerbation, despite their insufficient efficacy. Prior studies have shown that inhaled corticosteroid treatment lacks beneficial effects, such as anti-inflammatory and anti-protease functions in COPD patients [46,47]. Therefore, alternative and more effective pharmacological approaches, including anti-inflammatory and anti-protease effects, are needed for COPD therapy. In this direction, our current studies revealed that the extract of *E*. *pyrricholophum* attenuates lung inflammation and clinical manifestation in CSE-induced in vitro and in vivo COPD models. In summary, EPE suppresses macrophage, neutrophil, and lymphocyte recruitment and clinical symptoms, including elastic fiber disruption, increased bronchiole thickness, and mucus hypersecretion in PPE- and CSE-induced COPD mice. Our in vitro analyses showed that EPE repressed IL6 and IL8 expression by suppressing MAPKs and NFκB activity. Furthermore, our observations identified the anti-inflammatory function of corilagin, which was identified as a phytochemical constituent of EPE. The ellagitannin polyphenol compound corilagin is characterized by diverse pharmacological properties, including anti-inflammatory, antioxidative, anti-tumor, and anti-microbial effects. Notably, corilagin suppressed IL8 expression by inhibiting NFκB binding activity in human bronchial epithelial IB3-1 cells [48]. A recent study showed that corilagin attenuates bleomycin-induced fibrogenesis and tumor metastasis by inhibiting tumor growth factor β1-induced epithelial-mesenchymal transition in the presence of active lysyl oxidase-like 2 [49]. In addition, corilagin inhibits serine protease activity, such as prolyl endopeptidase, trypsin, and elastase, and shows potent ROS-scavenging activity [50]. In this context, our current studies suggested that corilagin and EPE may provide beneficial clinical effects for the treatment of chronic inflammatory diseases including, but not limited to, asthma, COPD, ulcerative colitis, and psoriasis. Further studies are needed to determine the molecular mechanisms associated with innate immune cell trafficking and elastolytic enzyme function in COPD pathogenesis at the cellular and transcriptional levels.

## Figures and Tables

**Figure 1 foods-10-02929-f001:**
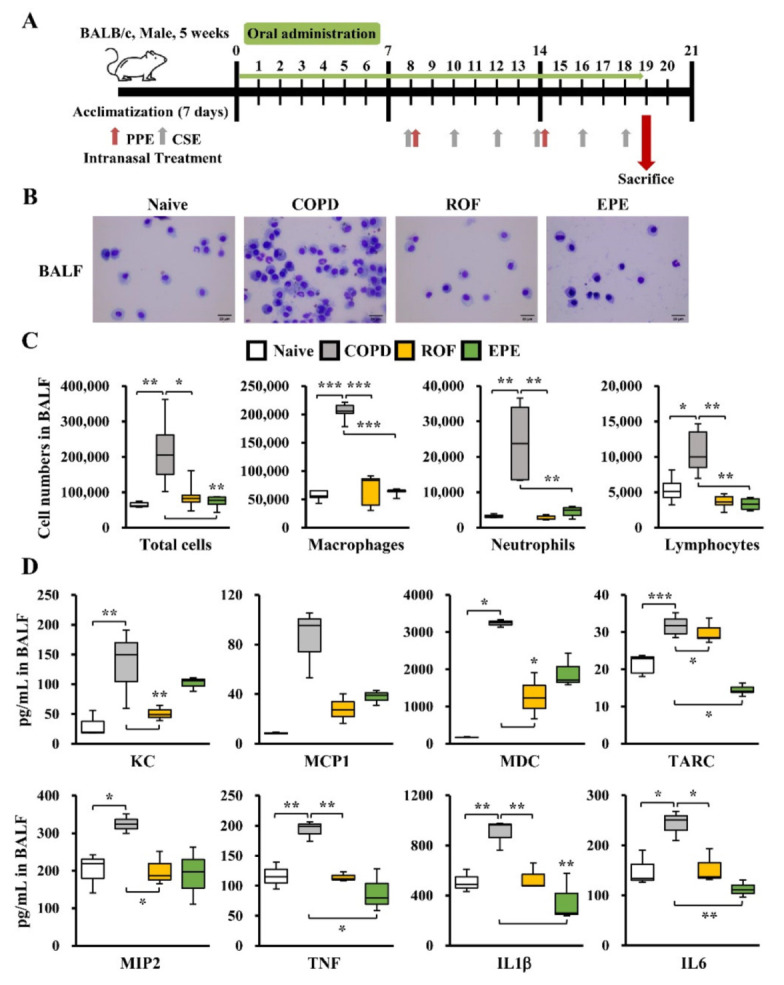
EPE represses inflammatory cell recruitments and pro-inflammatory gene expression in BALF. (**A**) Schematic representation of the experimental design of porcine pancreatic elastase (PPE) and cigarette smoke extract (CSE)-induced chronic obstructive pulmonary disease (COPD) mouse model. (**B**) The bronchoalveolar lavage fluid (BALF) was obtained from mice on day 19 after intranasal delivery of PPE and CSE. The neutrophil recruitments within the BALF were determined by Diff Quick staining, respectively (*n* = 10 per group). (**C**) Total cell, macrophages, neutrophils, and lymphocytes numbers in BALF following PPE and CSE exposure (*n* = 10). (**D**) BALF levels of KC, MCP1, MDC, TARC, MIP2, TNF, IL1β, and IL6 were analyzed by Q-plex assay kit (*n* = 10). Data were analyzed by one-way ANOVA. All values are reported as means SD. Scale bars, 20 μm. * *p* < 0.05, ** *p* < 0.01, *** *p* < 0.001.

**Figure 2 foods-10-02929-f002:**
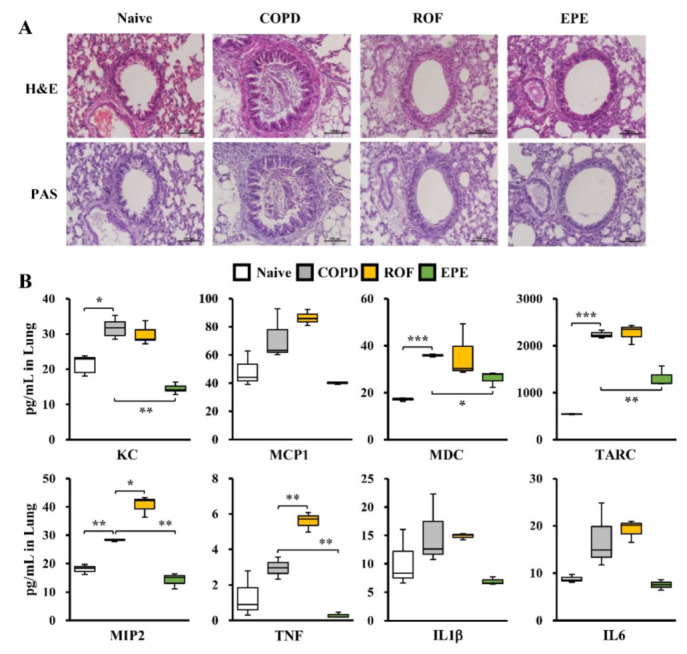
EPE attenuates lung inflammation in PPE and CSE-elicited COPD mice. (**A**) Mice were exposed to PPE and CSE intranasally in the presence or absence of *Epilobium pyrricholophum* extract (EPE) or roflumilast (ROF) treatment (*n* = 10 per group). On day 19, lung tissue sections of each mouse group were stained for hematoxylin and eosin and periodic acid-Schiff to quantify COPD-like histopathological symptoms and mucus production (*n* = 10). (**B**) Protein was extracted from whole lung tissues and measured for KC, MCP1, MDC, TARC, MIP2, TNF, IL1β, and IL6 levels by Q-plex assay kit (*n* = 10). Data were analyzed by one-way ANOVA. All values are reported as means SD. Scale bars, 100 μm. * *p* < 0.05, ** *p* < 0.01, *** *p* < 0.001.

**Figure 3 foods-10-02929-f003:**
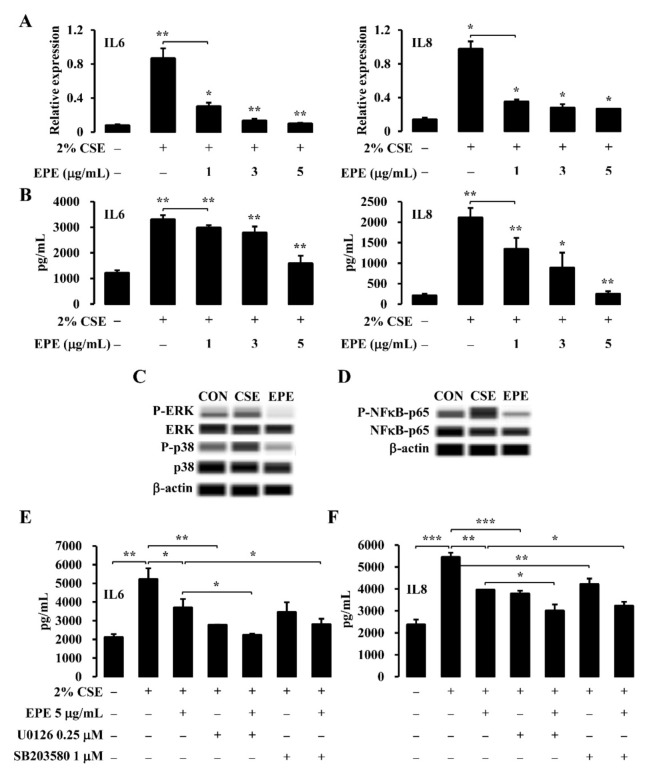
Following CSE exposure, EPE suppresses inflammatory gene expression through the MAPK and NFκB pathway in NCI-H292 cells. (**A**-**F**) NCI-H292 cells were treated with EPE following 2% CSE stimulation. Total RNA and protein samples were analyzed for IL6 and IL8 mRNA (**A**) and protein (**B**) expression by RT-qPCR and ELISA, respectively (*n* = 5). NCI-H292 cells were treated EPE in the presence or absence of pharmacological ERK (U0126) and p38 (SB203580) inhibitors following 2% CSE exposure. Total protein extracts from these studies were evaluated for p-ERK, total ERK, p-p38, total p38 (**C**), p-NFκB-P65, and total NFκB-p65 (**D**) levels by Western blot. β-actin was used as a loading control (*n* = 5). NCI-H292 cells were co-treated with EPE with either U0126 or SB203580 following 2% CSE stimulation. Protein levels were analyzed for IL6 (**E**) and IL8 (**F**) by ELISA analyses (*n* = 5). These experiments were performed three independent times. All values are reported as mean ± SD. * *p* < 0.05, ** *p* < 0.01, *** *p* < 0.001.

**Figure 4 foods-10-02929-f004:**
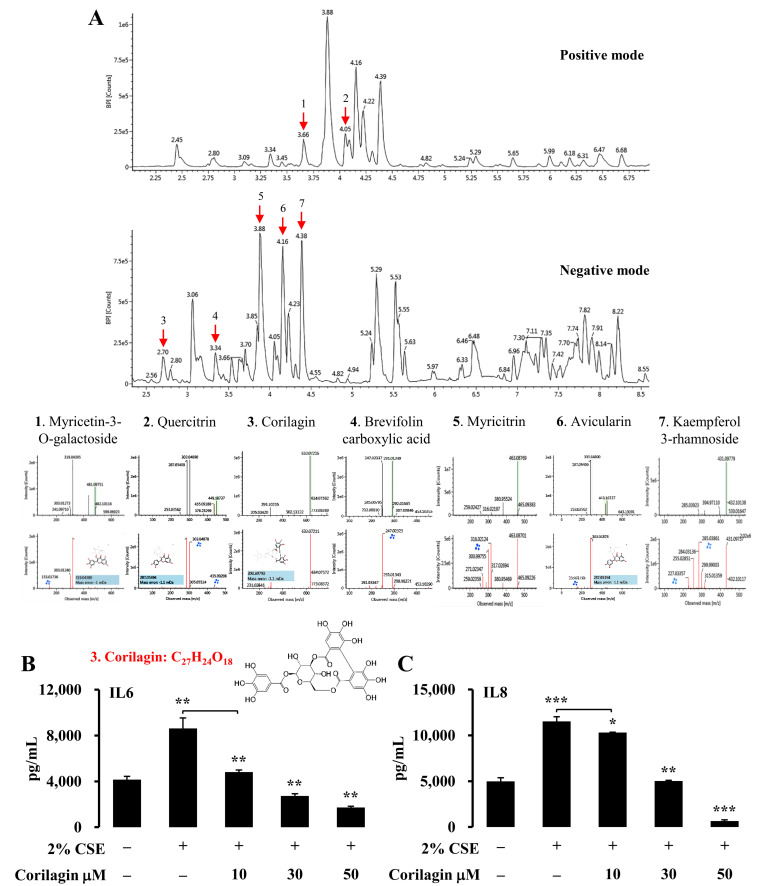
Identification of the active compounds in EPE that ameliorate inflammatory gene expression. (**A**) The mass chromatogram and MS/MS compounds data were obtained using UPLC-Q-TOF-MS with positive and negative mode. The MS spectra, chemical structure, and fragmentation ions of the seven identified compounds are shown. (**B** and **C**) NCI-H292 cells were treated with selective identified compounds following 2% CSE stimulation. Protein levels were analyzed for IL6 (**B**) and IL8 (**C**) by ELISA analyses (*n* = 5). These experiments were performed three independent times. All values are reported as mean ± SD. * *p* < 0.05, ** *p* < 0.01, *** *p* < 0.001.

**Table 1 foods-10-02929-t001:** Identification of bioactive compounds from EPE using UPLC-Q-TOF-MS analyses.

Peak No.	RT (min)	Measured Mass (*m*/*z*) (M + H)/(M − H)	Fragment Ions	Tentative Identification	Exact Mass	Formula
1	3.66	481.0975	153, 303, 319	Myricetin 3-O-galactoside	480.09	C_21_H_20_O_13_
2	4.05	447.0926	287, 303, 305, 435	Quercitrin	448.10	C_21_H_20_O_11_
3	2.70	633.0722	231, 301, 633, 634	Corilagin	634.08	C_27_H_22_O_18_
4	3.34	291.0135	191, 247, 291	Brevifolin carboxylic acid	292.02	C_13_H_8_O_8_
5	3.88	463.0877	259, 271, 301, 316	Myricitrin	464.09	C_21_H_20_O_12_
6	4.16	449.1074	153, 303, 287	Avicularin	434.08	C_20_H_18_O_11_
7	4.38	431.0978	227, 255, 284, 285	Kaempferol 3-rhamnoside	432.10	C_21_H_20_O_10_

**Table 2 foods-10-02929-t002:** Inhibitory effects of selective identified compounds from EPE on CSE-induced IL6 expression in NCI-H292 cells.

Selected Identified Compounds	IL6 (pg/mL)				
	Naive	CSE	10 mM	30 mM	50 mM
Myricetin 3-O-galactoside	4337 ± 540.55	7952 ± 294.25 ***	7241 ± 984.36	6170 ± 789.79 *	5957 ± 475.03 **
Quercitrin	1910 ± 215.29	5566 ± 550.26 **	5200 ± 323.57	6006 ± 66.02	5837 ± 183.59
Corilagin	4157 ± 314.88	8624 ± 1030.07 ***	4811 ± 207.17 **	2733 ± 213.61 **	1722 ± 121.74 **
Brevifolin carboxylic acid	3697 ± 308.24	7687 ± 612.90 ***	6129 ± 1004.49	5894 ± 1136.34	4856 ± 232.14 **
Myricitrin	2116 ± 734.62	6250 ± 1038.64 **	6433 ± 588.96	5559 ± 552.10	5721 ± 958.47
Avicularin	4039 ± 86.99	8405 ± 721.72 ***	6996 ± 1113.98	6455 ± 1193.42 *	5048 ± 130.94 **
Kaempferol 3-rhamnoside	3269 ± 14.01	7631 ± 438.60 **	7907 ± 582.67	8100 ± 196.39	8041 ± 731.14

All values are reported as mean ± SD. * *p* < 0.05, ** *p* < 0.01, *** *p* < 0.001.

**Table 3 foods-10-02929-t003:** Inhibitory effects of identified compounds from EPE on IL8 expression following CSE exposure in NCI-H292 cells.

Selected Identified Compounds	IL8 (pg/mL)				
	Naive	CSE	10 mM	30 mM	50 mM
Myricetin 3-O-galactoside	4935 ± 140.33	10307 ± 147.22 ***	10634 ± 304.01	9298 ± 67.32 **	9505 ± 643.80
Quercitrin	1632 ± 15.25	7140 ± 152.04 ***	7797 ± 243.46	8090 ± 20.78	7651 ± 217.29
Corilagin	4982 ± 449.37	11525 ± 572.59 **	10315 ± 27.27 *	5011 ± 98.98 **	655 ± 138.62 **
Brevifolin carboxylic acid	4680 ± 121.96	9770 ± 325.39 **	10042 ± 115.64	9605 ± 545.27	10149 ± 182.90
Myricitrin	1652 ± 311.55	7315 ± 376.04 ***	7769 ± 588.49	7213 ± 436.87	7404 ± 279.26
Avicularin	4643 ± 576.67	10948 ± 498.87 **	10748 ± 898.44	10692 ± 537.91	11149 ± 512.04
Kaempferol 3-rhamnoside	2452 ± 218.74	8071 ± 177.38 ***	8793 ± 329.75	9093 ± 422.45	8930 ± 141.12

All values are reported as mean ± SD. * *p* < 0.05, ** *p* < 0.01, *** *p* < 0.001.

## Data Availability

The data presented in this study are available upon request from the corresponding author.

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
