# Peer review of "Epilobiumpyrricholophum Extract Suppresses Porcine Pancreatic Elastase and Cigarette Smoke Extract-Induced Inflammatory response in a Chronic Obstructive Pulmonary Disease Model"

_foods, 2021, doi:10.3390/foods10122929_

Round 1

Reviewer 1 Report

This article describes the protective effects of Epilobium pyrricholophum extract (EPE) against Chronic Obstructive Pulmonary Disease (COPD) induced by cigarette smoke extract. The experimental design and the results seem to be exquisite and valid.

However, the following point would be considered to improve this article.

Minor point

Corilagin was reported to inhibit epithelial-mesenchymal transition induced by TGF-β1 and weaken fibrosis , for example, as shown below.

Wei, Y. ; Kim, T. J. ; Peng, D. H. ; Duan, D. ; Gibbons, D. L. ; Yamauchi, M. ; Jackson, J. R. ; Le Saux, C. J. ; Calhoun, C. ; Peters, J. ; Derynck, R. ; Backes, B. J. ; Chapman, H. A. Fibroblast-specific inhibition of TGF-β1 signaling attenuates lung and tumor fibrosis. J Clin Invest 2017, 127(10), 3675–3688. doi : 10.1172/JCI94624.

COPD occurs in combination with lung diseases, such as bronchial asthma, lung cancer, pulmonary fibrosis and interstitial pneumonia. As smoking is related to onset of these diseases, these diseases considered to be pulmonary complications that common onset factors are involved with COPD.

Therefore, the discussion may be enriched by the addition of the description of epithelial-mesenchymal transition, if authors don't have any plans to investigate the effects of corilagin on epithelial-mesenchymal transition in this model at this point in time.

Author Response

Answer: We agree with the reviewer on this comment. Therefore, we have revised the text in the discussion section with appropriate reference (page13, line 470-473, A recent study showed that corilagin attenuates bleomycin-induced fibrogenesis and tumor metastasis by inhibiting tumor growth factor b1-induced epithelial-mesenchymal transition in the presence of active lysyl oxidase-like 2).

Reviewer 2 Report

Jung et al. presented in their manuscript anti-inflammatory effects of an Epilobium extract in a mouse model of COPD.  COPD is a chronic inflammatory lung disease with limited therapeutic options. Therefore the rationality to identify new drugs for COPD treatment is given.  In the COPD model the authors show nicely EPE-mediated reduction of immune cell infiltration and inflammatory cytokine and chemokine expression in BALF. These results are supported by histopathological evaluation of lung tissue in EPE-treated COPD mice. The postulated mode of action of EPE is inhibition of p38MAPK, ERK or NF-κB, a mechanism often addressed by natural compounds.  One plus is that the authors try to identify the bioactive compound of EPE extract that could be responsible for the observed effects. Some points have to be addressed.

1) Abstract Line 34-35: The study neither showed attenuated ROS production nor reduced myeloperoxidase  expression in murine or human innate immune cells. In addition, reduced expression of inflammatory cytokines was shown in BALF and a mucoepidermoid cell line and not in innate immune cells. Please correct this statement.

2) EPE administration started at day 0. Therefore it is a rather a prophylactic than a therapeutic treatment. This should be mentioned.

3) There is a discrepancy between figure 2A and 2B concerning the ROF-treated control group. Histopathological evaluation implies less severe disease symptoms in this group, but this result is not reflected by the protein data. Compared to the EPE-group, no reduction (sometimes even an increase) of inflammatory gene expression in the lung tissue was detected in the ROF group. This conflicting result is not discussed at all by the authors.

4) The authors should add the primer sequence used to detect IL-6 and IL-8 mRNA expression. As relative mRNA expression is shown in Figure 3A, authors should add information about the housekeeping gene used in their qRT-PCR analyses.

5) The western blots presented in Figure 3 C and D are cropped too much.

6) Figure 3E: Is the inhibition of IL-6 production by SB 203580 significant? What is the rationality to use a combination of EPE and U0126 or EPE and SB 203580? The authors neither describe the results of this combination nor discuss them.

7) Line 385: No experimental evidences  are presented that the effect is specific for innate immune cells.

8) Discussion line 405-415 and 429-434: The discussion about NET, α1AT-1 does not fit to the results presented in the manuscript.

Author Response

1. Answer: We really appreciate the reviewer on this comment. Therefore, we have modified the text in the abstract section (page 1, line, 34-35, Recent studies showed that Epilobium species attenuated ROS, myeloperoxidase, and inflammatory cytokine production in murine and human innate immune cells).

2. Answer: We appreciate the reviewer for this insight. Therefore, we have modified the text in the results section (page 5, line 211-212, Accordingly, oral administration of EPE was initiated 7 days before PPE and CSE induction to identify both prophylactic and therapeutic effects).

3. Answer: We really appreciate the reviewer on this comment. PDE4 inhibitors are currently used as potential therapies for diseases associated with airway inflammation. ROF treatment reduces infiltration of neutrophils and inflammatory gene expression in BALF, obviously. However, PDE4 inhibitors have been shown to cause an unknown etiology in lung tissues of in vivo model. McCluskie et al1. showed that 100 mg/kg of roflumilast and piclamilast caused a significant increase in plasma and lung tissue KC levels and the number of neutrophils. In addition, piclamilast and roflumilast induced increases of IL8 release from HUVECs in vitro. Kasetty et al2. also identified that roflumilast decreases the number of neutrophils in BALF but increases the number in lung tissues. Furthermore, the roflumilast elevated plasma levels of IL6, MCP1, and TNF after bacterial infection in the mouse model. We assumed that unexpected effects of PDE4 inhibitors may appear due to the complexity of the interaction between tissues and cells, and we are conducting several studies to discover the mechanisms involved, thus this will be discussed in future research. I wish our response will be helpful to your insightful comments.

Ref 1. J Pharmacol Exp Ther. 2006 Oct;319(1):468-76. doi: 10.1124/jpet.106.105080. Epub 2006 Jul 21.

Ref 2. J Pharmacol Exp Ther. 2016 Apr;357(1):66-72. doi: 10.1124/jpet.115.229641. Epub 2016 Feb 10.

4. Answer: We really appreciate the reviewer on this comment. Therefore, we have modified the text in the materials and methods section (page 4, line 167-171, The primer sequences were as follows: IL6 forward, 5′-ATGAACTCCTTCTCCACAAGC-3′ and reverse, 5′-TGGACTGCAGGAACTCCTT-3′; IL8 forward, 5′-ATGACTTCCAAGCTGGCCGTGGCT-3′ and reverse, 5′-TCTCAGCCCTCTTCAAAAACTTCTC-3′; GAPDH forward, 5′-TGCACCACCAACTGCTTA-3′ and reverse, 5′-GGCATGGACTGTGGTCAT-3′).

5. Answer: We appreciate the reviewer on this comment. In the present study, all the WB data were processed using an automated capillary-based size sorting system (WES machine) and linked automated software. Therefore, sometimes produced images have insufficient space or resolution. Although we tried to make the figure better following your suggestion, we couldn’t enhance the quality of Figures 3C and 3D. If we confirm all the WB work using a manual system, it would be better. However, we don’t have enough time for the revision of this manuscript. I wish our response will be helpful to your comments.

6. Answer: We appreciate the reviewer for this insight. We already discussed the correlation of EPE with specific pharmacological ERK and p38 inhibitors, U0126 and SB203580 in the result section. EPE significantly suppressed CSE-induced IL6 and IL8 mRNA and protein expression in NCI-H292 cells. In addition, it has been well known that CSE-induced IL6 and IL8 expressions were dependent on the phosphorylation of MAPKs. Therefore, to investigate the signaling pathway involved in CSE-induced IL6 and IL8 production in NCI-H292 cells, we evaluated the phosphorylation levels of ERK, p38, and JNK (data not shown) by western blotting. Our observations identified that EPE suppresses CSE-induced IL6 and IL8 production through the inhibition of ERK and p38 phosphorylation (JNK was not working) in NCI-H292 cells. We checked all the significance levels using *p<0.05 and **p<0.01 in all groups. Because of the p-value between CSE+EPE and CSE+EPE+SB203580 was 0.06, we couldn`t label it in figure 3E. 

7. Answer: We appreciate the reviewer for this comment. Actually, our in vivo data showed that EPE suppresses the infiltration of macrophages and neutrophils into the BALF and lungs after PPE and CSE exposure. These cells play important roles in innate immunity. However, there are no specific molecular mechanisms or correlation with EPE and innate immune cells in this study. Therefore, we have modified the text in the discussion section (page 12, line 384-386, Collectively, our observations demonstrate that EPE suppresses the immune response and inflammatory gene expression in a murine COPD model).

8. Answer: We agree with the reviewer on this comment. Therefore, we have removed the text in the discussion section (page 13, line 429-431, Intriguingly, elevated levels of MMP9 in sputum and plasma induced emphysema as well as diminished FEV1, carbon monoxide transfer factor, and oxygen saturation in COPD exacerbation patients exposed to CS). However, emphasizing an explanation about the importance of neutrophils and their functions under COPD pathogenesis, we remained to discuss NET and α1AT-1 in the discussion section, lines 405-415.

Reviewer 3 Report

The authors present a work entitled "Epilobium pyrricholophum Extract Suppresses Porcine Pancreatic Elastase and Cigarette Smoke Extract-Induced Inflammatory response in a Chronic Obstructive Pulmonary Disease Model". Although the work presented is interesting, there are several critical points that the authors should consider.

Below are some suggestions / corrections to consider in order to improve the overall quality of the work.

This Manuscript is submitted to the Special Issue “Functional Food and Bioactive Food Components”, Section Nutraceuticals and Functional Foods. Please, report in introduction and discussion some references regarding the current or traditional use of these species in food or nutraceutical fields.

In the abstract at Ln 34-36 “The present study showed that Epilobium species attenuated ROS, myeloperoxidase, and inflammatory cytokine production in murine and human innate immune cells” is reported, and at Ln 38 “In this study, Epilobium pyrricholophum extract…”. Please, clarify if the reported activity is ascribed to the generic Epilobium species or only to

Epilobium pyrricholophum.

Ln 65: “The present study…” which one? That cited in the reference? Please clarify this statement.

Ln 80-81: “This study extracted E. pyrricholophum belonging to the Onagraceae family distributed in Asian countries, including Korea, China, and Japan, using methanol”, please organize this sentence better

Ln 120: “NCI-H292 cells…”. Please, complete the cell line description such as “NCI-H292 mucoepidermoid carcinoma cells from human lungs”

The results should be expressed more clearly. In the abstract and in the introduction the authors declare that they use Epilobium extract and the results refer to his action. In the results and discussion there is talk of pharmacological action due to compounds present in the extract. Please make the discussion more homogeneous and introduce in the abstract and introduction the pharmacological action due to the isolated compounds. 

Author Response

1. Answer: We appreciate the reviewer for this insight. We already referred qualified review paper discussing, physiological function, phytochemical constitution, and usage of Epilobium species (Ref. 9) in this manuscript. We have referred one more recent review paper in the introduction section following your kind suggestion (page 2, line 74-77, Ref 10. Phytother Res. 2018 Jul;32(7):1229-1240. doi: 10.1002/ptr.6072. Epub 2018 Mar 25.).

2. Answer: We really appreciate the reviewer on this comment. Therefore, we have modified the text in the abstract section (page 1, line 34-35, Recent studies showed that Epilobium species attenuated ROS, myeloperoxidase, and inflammatory cytokine production in murine and human innate immune cells).

3. Answer: We really appreciate the reviewer on this comment. Therefore, we have modified the text in the introduction section (page 2, line 65, A recent study showed that Epilobium species, including E. angustifolium, E. hirsutum, and E. parviflorum suppress myeloperoxidase, leukotriene B4, and ROS production in human neutrophils).

4. Answer: We appreciate the reviewer for this insight. Therefore, we have modified the text in the introduction section (page 2, line 80-83, In this study, we analyzed the phytochemical composition of E. pyrricholophum belonging to the Onagraceae family distributed in Asian countries including Korea, China, and Japan, using ultra-performance liquid chromatography quadrupole time of flight (UPLC-Q-TOF) MS).

5. Answer: We really appreciate the reviewer on this comment. Therefore, we have modified the text in the materials and methods section (page 3, line 119, NCI-H292 human airway epithelial cells were cultured in RPMI 1640 medium).

6. Answer: We appreciate the reviewer for this insight. Therefore, we have modified the text in the previous manuscript with appropriate references following the reviewer`s suggestions.

Round 2

Reviewer 2 Report

The authors addressed my questions. Nevertheless, some points have to be considered.

Point 3 (Figure 2A+B)

Thanks for the information about rofumilast. I would appreciate, when they would add some of these to the discussion.

Point 6/Figure 3E + 3F

The authors still do not explain the results or the benefit of the combination of  CSE +EPE + U0126  (column 5) or CSE + EPE + SB 203580 (column 7) on IL6 and IL8 production.  In line 309-313 they describe the effect of CSE (column 2), CSE +EPE (column 3), CSE + U0126  (column 4),  and CSE + SB 203580 (column 6):

“As anticipated, CSE-induced IL6 and IL8 productions were significantly downregulated by U0126 and SB203580 treatment (Figure 3E and 3F). Remarkably, EPE treatment significantly attenuated IL6 and IL8 production in NCI-H292 cells following CSE exposure, similar to the pharmacological blockade of the ERK and p38 pathways (Figure 3E and 3F). “

They should add some more information.

Author Response

1. Answer: We agree with the reviewer on this comment. In context, it seems better to discuss ROF in the result section 2, therefore, we have revised the text in the result section with appropriate references (page 7, line 258-271, However, mice that received EPE exhibited significantly suppressed inflammatory gene expression in the lung tissues following PPE and CSE exposure, consistent with the observations in BALF and immunohistochemistry analyses (Figure 2B). ROF treatment reduced PPE and CSE-induced KC and IL1b levels in the lungs whereas levels of MCP1, MDC, TARC, MIP2, TNF, and IL6 were increased, unexpectedly (Figure 2B). Although phosphodiesterase 4 inhibitors are used as a therapeutic application for airway inflammation, ROF has been shown to cause an unknown etiology in experimental models. Notably, 100 mg/kg ROF induced increases in the number of neutrophils and KC levels in plasma and the lung tissues of the mice model [18]. Furthermore, Kasetty et al. [19] demonstrated that ROF increases plasma levels of IL6, MCP1, and TNF while suppressing the infiltration of neutrophils in the BALF after bacterial infection, similar to our observations. Therefore, we are conducting further studies to investigate the appropriate dosage and molecular mechanisms involved that ROF treatment in the elastolytic enzyme and CSE-induced COPD model).

2. Answer: We really appreciate the reviewer on this comment. Therefore, we have modified the text in the result section (page 8, line, 316-319, In addition, the combination of EPE and either U0126 or SB20358 treatment exhibited significantly lower levels of IL6 and IL8 compared to a single treatment of EPE or each inhibitor in NCI-H292 cells following CSE exposure (Figure 3E and 3F)). 

Reviewer 3 Report

The authors have answered in a comprehensive manner to the comments / suggestions previously sent. In the present version the manuscript seems much improved although the authors still do not justify the coherence of the manuscript with the themes of the journal and of the Special Issue in particular.

Author Response

1. Answer: We really appreciate the reviewer on this comment. Therefore, we have modified the text in the introduction and discussion section (page 2, line, 83-84, Notably, corilagin, which was identified as a phytochemical constituent of E. pyrricholophum significantly attenuated CSE-induced IL6 and IL8 expression. page 13, line 478, Furthermore, our observations identified the anti-inflammatory function of corilagin, which was identified as a phytochemical constituent of EPE).